# A brief, theory-driven patient education video reduces high-risk over-the-counter nonsteroidal anti-inflammatory drug (NSAID) use

Matthew G. Vinson[1], Allison N. Tegge[2,3], Mamata R. Tokala [4], Cara R. Spivey[1,5], Martha M. Tenzer[4], Jeffrey S. Stein[2], John W. Epling[1,6], Michelle S. Rockwell [1,6]*

1 Virginia Tech Carilion School of Medicine, Roanoke, Virginia, United States of America, 2 Fralin Biomedical Research Institute at VTC, Roanoke, Virginia, United States of America, 3 Addiction Recovery Research Center, Roanoke, Virginia, United States of America, 4 Health Analytics Research Team, Carilion Clinic, Roanoke, Virginia, United States of America, 5 Research and Development, Carilion Clinic, Roanoke, Virginia, United States of America, 6 Department of Family and Community Medicine, Carilion Clinic, Roanoke, Virginia, United States of America

* msrock@vt.edu

## Abstract

Professional guidelines advise against regular or long-term NSAID use in most patients with chronic kidney disease (CKD), heart failure (HF), and hypertension (HTN) due to risk of adverse events. Nevertheless, over-the-counter (OTC) NSAIDs are broadly accessible and frequently used among this population. Efforts to decrease high-risk OTC NSAID use have the potential to improve safety and reduce chronic disease burden. This randomized controlled trial evaluated the effectiveness of a brief, electronically-administered educational video in reducing high-risk OTC NSAID use. Adult participants with CKD, HF, and/or HTN who self-identified as regular NSAID users (≥3 times/week for 3 months) were invited to participate. Participants (n = 425) were randomized to either view an electronically-administered educational video informed by the COM-B behavioral change model (VIDEO, n = 223) or the FDA Drug Facts label for NSAIDs (CONTROL, n = 202). Intent to decrease OTC NSAIDs was evaluated via 11-point contemplation ladder immediately and 4 weeks post-intervention, with self-reported NSAID Exposure assessed at 4 weeks. We also evaluated current and recent pain levels at baseline and 4 weeks. Intent to decrease OTC NSAID use (4.28 (SD: 3.45) ladder rungs) and NSAID exposure (20.14 (SD: 13.66) dose-days per month) did not differ between groups at baseline. Intent to decrease OTC NSAID use increased more from baseline to immediately post-intervention in VIDEO vs. CONTROL (1.32 (SD: 2.80) vs. 0.55 (SD: 1.99) rungs, p < 0.001), with greater improvements for those with lower baseline intent. VIDEO and CONTROL were associated with a similar rise in intent to decrease OTC NSAID use (1.92 (SD: 4.41) vs. 1.36 (SD: 3.46), p = 0.150) and a similar decrease in NSAIDs exposure (−32.8% in VIDEO and −36.5% in CONTROL, p = 0.520) 4 weeks

**Data availability statement:** De-identified All study data are available on OSF using the following link: https://osf.io/e2utg/files/u7ztj.

**Funding:** This research was funded, in part, by a collaborative grant from the Virginia Tech Libraries. The funders had no role in study design, data collection and analysis, decision to publish, or preparation of the manuscript.

**Competing interests:** The authors have declared that no competing interests exist.

post-intervention. Pain levels did not differ between groups. Results suggest that a low-burden, electronically-administered intervention reduce high-risk medication use among patients with CKD, HF, and/or HTN.

## Introduction

Over-the-counter (OTC) medications, which can be purchased without a prescription, are broadly available in retail pharmacies, supermarkets and convenience stores, online retailers, and other outlets throughout the United States (US). Although commonly used and generally considered safe, OTC medications can increase the risk for harm in some individuals [1]. Nonsteroidal anti-inflammatory drugs (NSAIDs), among the most commonly used OTC medications, are effective in treating pain, inflammation, and fever but can also induce adverse gastrointestinal, renal, and cardiovascular effects [2–7]. Professional organizations recommend that patients with chronic health conditions such as chronic kidney disease (CKD), heart failure (HF), and hypertension (HTN) limit or avoid the regular use of NSAIDs [8–10]. Despite these recommendations, regular NSAID use continues among individuals with CKD, HF, and HTN at undesirable and increasing rates [5,6]. As more than 50% of US adults have CKD, HF, and/or HTN [11], population-level strategies to decrease high-risk OTC NSAID use are needed.

To date, documented interventions aimed at reducing utilization of potentially harmful or high-risk NSAIDs are primarily clinician-facing, focus on gastrointestinal risks rather than cardiovascular and renal risks, and largely address prescription NSAIDs without focus on OTC NSAIDs [12]. The few available public-facing interventions to reduce use of high-risk NSAIDs report mixed results. Review of a Food and Drug Administration (FDA) pamphlet about NSAID safety and risks resulted in no improvement in intent to decrease high-risk NSAIDs [13]. In contrast, Pai et al. [14] showed that 65% of CKD patients who viewed an NSAID safety educational handout or video reported intent to decrease or avoid high-risk NSAID behaviors. In addition, a multi-session nurse-administered education program was effective in improving knowledge and self-efficacy related to OTC NSAID-associated risks but was resource-intensive to implement [15]. Identification of an effective, low-burden intervention to de-implement high-risk OTC NSAID use has the potential to improve medication safety and population health outcomes.

In this study, we evaluated the effectiveness of a brief educational video in decreasing high-risk OTC NSAID use. Informed by the COM-B behavior change model [16], which emphasizes that effective behavior change requires an interaction between capacity, opportunity, and motivation domains, the video incorporated a combination of narrative content and authentic representations of real-world scenarios. We hypothesized that the educational video would be associated with a decrease in intent to use regular OTC NSAIDs and self-reported OTC NSAID use immediately and 4 weeks after viewing.

## Methods

### Study design

We performed a randomized controlled trial (RCT) in January to July 2023 to evaluate the effectiveness of a brief educational video intervention on high-risk OTC NSAID use in patients with CKD, HF, and/or HTN. Specifically, we evaluated the change in intent to decrease regular use of OTC NSAIDs immediately following the video intervention and 4 weeks after the intervention, in addition to the change in intent to decrease regular use of OTC NSAIDs 4 weeks after the intervention. We also explored secondary outcomes 4 weeks after the intervention: change in ratings of capability, opportunity, and motivation to decrease high-risk OTC NSAID use, and pain measures.

This study was approved by the Institutional Review Board of Carilion Clinic (IRB-22–1766) and is registered with clinicaltrials.gov (NCT06575205) . All participants provided informed consent before beginning the study.

### Setting and participants

Adult (≥18 years of age) patients from a large health system in the southeastern US with a diagnosis of CKD, HF, and/or HTN documented in the health system's electronic health record (EHR) 2 or more times during the previous 12 months were invited via email to complete an electronic screening survey to determine study eligibility. Those who self-identified as regular OTC NSAID users (i.e., high-risk OTC NSAID users) on the screening survey (S1 File) were invited to participate in the trial. *Regular use* was defined as self-reported use of any type or dose of OTC NSAID 3 or more days per week during the past 3 months [6]. Diagnosis codes used to identify prospective participants are shown in S2 File. The screening instrument and trial were administered electronically using Qualtrics (Provo, UT).

### Procedures

High-risk OTC NSAID-users who consented to participate were randomized to 1 of 2 groups: educational video intervention (VIDEO) or control (CONTROL) using an automated randomization process described in S3 File. Following the completion of a pre-intervention (baseline) assessment, VIDEO participants viewed the brief educational video and CONTROL participants viewed a still image of the FDA Drug Facts label for OTC NSAIDs [17] on their own computer or mobile device. The video, which reflects an original creation by our team, can be accessed online at https://www.youtube.com/watch?v=zsiJjXk162E (S4 File).The label image is shown in S5 File.

Participants from both groups completed a post-intervention assessment (immediately following viewing of the video or label) and a follow-up assessment (4 weeks after viewing the video or label). All assessments were administered via REDCap (version 13.9.2, Nashville, TN). Participants were compensated with a gift card at the completion of the follow-up assessment. **Fig 1** provides an overview of the study procedures.

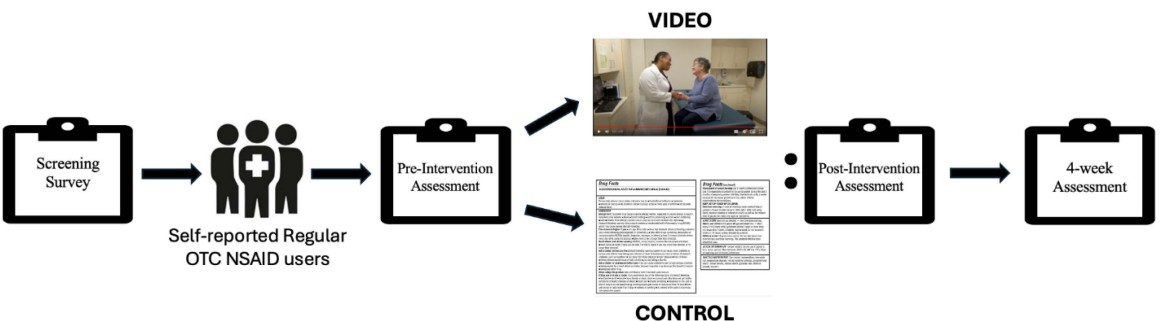

**Fig 1. Overview of study procedures.**

Based on a priori sample size calculation, 1054 study participants were needed to achieve a small effect size and statistical significance at p < 0.05, assuming a 50% response rate to the 4-week follow-up assessment survey. To account for non-completers and attrition, we aimed to recruit 1100 high-risk OTC NSAID users.

## Video intervention

The VIDEO intervention consisted of a novel, 4-minute educational video designed and produced by the research team in collaboration with content, curricular design, and video communications experts. The video script was written at <8th grade reading level. The structure and content of the video were guided by the COM-B behavior change model [16] and the work of Abu Abed et al., [18] who recommend that patient education interventions incorporate a narrative format telling a relatable story or feature authentic representations of applicable scenarios. According to COM-B, effective behavior change requires an interaction between three domains: knowledge and capacity (Capability), external forces including available resources and support (Opportunity), and willingness or desire (Motivation) to change Behavior.

The video begins with a narrator defining OTC NSAIDs and the potential harms of NSAIDs (Capability, Motivation). Next, physician and patient actors discuss high-risk NSAIDs, including reasons to avoid regular use (Motivation) and potential alternatives (Opportunity). The narrator concludes with a recommendation that patients speak with their healthcare provider for personalized recommendations (Opportunity).

Prior to the trial, the video was reviewed, pilot tested, and iteratively refined based on feedback from community members, clinicians, educators, and public health professionals.

## Assessments

Pre-intervention and immediate post-intervention assessments were comprised of questions about general demographics, intent to decrease regular use of OTC NSAIDs, capability, opportunity, and motivation to decrease regular use of OTC NSAIDs, and pain measures. The 4-week post-intervention follow-up assessment included questions about intent to decrease regular use of OTC NSAIDs, self-reported OTC NSAID use during the previous 4 weeks, capability, opportunity, and motivation to decrease regular use of OTC NSAIDs, and pain measures. All assessments are included in S6 File.

Intent to decrease use of OTC NSAIDs was assessed via the 11-point contemplation ladder, informed by the work of Biener and Adams [19,20], which aligns with the Transtheoretical Theory Model of Change [21,22] and has previously been validated for assessing readiness to change a variety of health behaviors [23–26]. Participants were asked to select the rung of the ladder corresponding to their current intention to decrease their use of OTC NSAIDs. Ladder rungs ranged from 0 = "*No thoughts about decreasing my use of over-the-counter NSAIDs*" to 10 = "*I have decided to decrease my use of over-the-counter NSAIDs and will never use them the same way again.*"

Self-reported OTC NSAID use was evaluated using 4 questions informed by National Health Interview Survey [27] and National Health and Nutrition Examination Survey [5]. Participants were asked to select how often they have taken OTC ibuprofen or naproxen, aspirin, headache tablets and powders, and other OTC NSAIDs such as some cold & sinus medications and some menstrual pain medications, with examples provided for each. We guided participants to distinguish aspirin used for pain through a statement advising that low-dose or baby aspirin taken for cardiovascular prevention should not be included in their response [28].

COM-B domains (capability, opportunity, and motivation) were assessed via 7 questions informed by the COM-B Questionnaire developed by Keyworth et al. [29]. Questions about current pain level, average pain level over the past 90 days (average recent pain), and worst pain level over the past 90 days (worst recent pain) were informed by the PROMIS Short Form v.1.0 Pain Intensity [30].

## Outcome measures

Primary outcomes included intent to decrease regular use of OTC NSAIDs (contemplation ladder score of 0–10) and self-reported OTC NSAID use (NSAID Exposure). OTC NSAID Exposure represents the summative volume of all OTC NSAID categories expressed as dose-days per month to account for the possibility of taking multiple versions of OTC NSAIDs. To calculate dose-days per month, we applied the following formula:

$$\text{Weekly NSAID (days per week)} = \text{NSAID A (days per week)} + \text{NSAID B (days per week)}$$
$$\text{NSAID C (days per week)} + \text{NSAID D (days per week)}$$

$$\text{Dose-Days per month} = \text{Weekly NSAID (days per week) X 4}$$

When the frequency response was a range (e.g., 3–5 or 1–2), we used an average value (e.g., 4 and 1.5, respectively). For example, if ibuprofen was taken 3–5 days per week and naproxen 1–2 days per week, the NSAID exposure would be averaged to (5.5 dose-days per week x 4 weeks) per month = NSAID Exposure of 22 dose-days per month.

Secondary outcomes included responses to 7 COM-B domain questions (11-point Likert scale, where 0 = *strongly disagree* and 10 = *strongly agree*) and 3 pain measures (5-point Likert scale for each: current pain, average recent pain, and worst recent pain, with 1 = *no pain* and 5 = *very severe pain*).

## Data analysis

Demographic and clinical characteristics were summarized using mean (standard deviations) and frequencies (percentages) and compared between the VIDEO and CONTROL groups using *t*-tests and chi-square tests, as appropriate. Group differences in intent to decrease OTC NSAID use, total NSAID Exposure, and the COM-B domains were compared at pre-intervention and post-intervention time points (immediate post-intervention and 4-week follow-up) using t-tests, as were changes in each score from pre- to post-intervention. Pain measures were evaluated similarly using a Wilcox rank sum test due to the non-parametric attributes of the pain measures (i.e., the 5-point Likert scale).

We evaluated the moderating role of pre-intervention (baseline) intent to decrease regular use of OTC NSAIDs, COM-B domains, and pain measures on post-intervention intent to decrease OTC NSAID use and changes in total OTC NSAID exposure using linear regression and including an interaction term between intervention group (VIDEO vs. CONTROL) and these moderators. We utilized Fisher's exact tests to examine potential associations between participants who did not complete the 4-week follow-up assessment and demographics, group assignment, and post-intervention intent to decrease regular OTC NSAID use. All analyses were conducted in R (Version 4.1.2) using a significance level of 0.05.

## Results

Out of 3233 individuals who screened for the study, 1452 (44.9%) initially self-identified as regular OTC NSAID users. After exclusions for ineligibility (see consort diagram in S7 File), 1320 (40.8%) remained. Of these, 917 participants consented and proceeded to the pre- and post-intervention assessment, with n = 466 (50.8%) randomized to VIDEO and n = 451 (49.2%) randomized to CONTROL. A total of n = 425 (46.3%) participants completed the 4-week follow-up assessment (VIDEO: n = 223 and CONTROL: n = 202). There were no significant differences in demographics and clinical characteristics of participants who completed the intervention and immediate post-intervention assessment and those who completed the 4-week follow-up assessment.

The overall cohort was 59.7 (SD: 12.1) years of age, 58.2% female, 90.7% White, 98.8% Non-Hispanic, and 43.1% insured by Medicare, with 41.4% having completed some college (Table 1). Demographics were representative of the patient population. Almost all patients had a diagnosis of HTN, while 22.7% and 18.6% were diagnosed with CKD or HF, respectively (Table 1). Demographic and clinical characteristics of the VIDEO and CONTROL groups were similar except for education level (Table 1).

## Primary outcomes

**Intent to decrease regular use of OTC NSAIDs immediately post-intervention.** At baseline, there was no significant difference between VIDEO and CONTROL in intent to decrease regular use of OTC NSAIDs (4.07 (SD: 3.56)

**Table 1. Demographic and clinical characteristics of the study cohort by group.**

| | Overall (n=917) | Pre-Intervention Assessment (Baseline) | | | 4-Week Post-Intervention Assessment (Follow-up) | | |
|---|---|---|---|---|---|---|---|
| | | VIDEO (n=466) | CONTROL (n=451) | P-value | VIDEO (n=223) | CONTROL (n=202) | P-value |
| **Age (years)** | 59.7±12.1 | 59.7±12.0 | 59.6±12.2 | 0.87 | 59.0±12.3 | 59.9±12.6 | 0.42 |
| **Biological Sex** | | | | | | | |
| Female | 534 (58.2%) | 262 (56.2%) | 272 (60.3%) | 0.24 | 124 (55.6%) | 130 (64.4%) | 0.08 |
| Male | 383 (41.8%) | 204 (43.8%) | 179 (39.7%) | | 99 (44.4%) | 72 (35.6%) | |
| **Race** | | | | | | | |
| Black/African American | 68 (7.4%) | 34 (7.3%) | 34 (7.5%) | 0.71 | 16 (7.2%) | 16 (7.9%) | 0.84 |
| White | 832 (90.7%) | 425 (91.2%) | 407 (90.2%) | | 204 (91.5%) | 182 (90.1%) | |
| Other/Refused to Answer | 17 (1.9%) | 7 (1.5%) | 10 (2.2%) | | 3 (1.3%) | 4 (2.0%) | |
| **Ethnicity** | | | | | | | |
| Hispanic | 6 (0.7%) | 4 (0.9%) | 2 (0.4%) | 0.66 | 2 (0.9%) | 0 (0%) | 0.40 |
| Non-Hispanic | 906 (98.8%) | 460 (98.7%) | 446 (98.9%) | | 220 (98.7%) | 201 (99.5%) | |
| Unknown | 5 (0.5%) | 2 (0.4%) | 3 (0.7%) | | 1 (0.4%) | 1 (0.5%) | |
| **Insurer** | | | | | | | |
| Commercial | 366 (39.9%) | 187 (40.1%) | 179 (39.7%) | 0.69 | 90 (40.4%) | 87 (43.1%) | 0.53 |
| Medicaid | 123 (13.4%) | 68 (14.6%) | 55 (12.2%) | | 34 (15.2%) | 22 (10.9%) | |
| Medicare | 395 (43.1%) | 194 (41.6%) | 201 (44.6%) | | 92 (41.3%) | 84 (41.6%) | |
| Other | 33 (3.6%) | 17 (3.6%) | 16 (3.5%) | | 7 (3.1%) | 9 (4.5%) | |
| **Education** | | | | | | | |
| High school or less | 201 (21.9%) | 97 (20.8%) | 104 (23.1%) | 0.03* | 40 (18.0%) | 35 (17.3%) | 0.46 |
| Some college | 380 (41.4%) | 186 (39.9%) | 194 (43.0%) | | 90 (40.5%) | 97 (48.0%) | |
| Bachelor's | 173 (18.9%) | 104 (22.3%) | 69 (15.3%) | | 52 (23.4%) | 34 (16.8%) | |
| Graduate level | 144 (15.7%) | 66 (14.2%) | 78 (17.3%) | | 39 (17.6%) | 35 (17.3%) | |
| Other/NA | 19 (2.1%) | 13 (2.8%) | 6 (1.3%) | | 1 (0.5%) | 1 (0.5%) | |
| **Diagnosis** | | | | | | | |
| CKD† –yes | 208 (22.7%) | 103 (22.1%) | 105 (23.3%) | 0.73 | 43 (19.3%) | 51 (23.2%) | 0.17 |
| HF‡ --yes | 171 (18.6%) | 88 (18.9%) | 83 (18.4%) | 0.92 | 37 (16.6) | 33 (16.3%) | 0.99 |
| HTN§ -- yes | 912 (99.5%) | 463 (99.4%) | 449 (99.6%) | 0.99 | 221 (99.1%) | 201 (99.5%) | 0.99 |

*p<0.05

† CKD=chronic kidney disease

‡ HF=heart failure

§ HTN=hypertension

vs. 4.23 (SD: 3.50) rungs, respectively, p = 0.51) At the immediate post-intervention assessment, intent to decrease regular use of OTC NSAIDs was significantly higher in VIDEO compared with CONTROL (5.39 (SD: 3.49) vs. 4.78 (SD: 3.49) rungs, respectively, p = 0.008). The absolute change in intent to decrease regular use of OTC NSAIDs was greater in VIDEO than CONTROL (1.32 (SD: 2.80) vs. 0.55 (SD: 1.99) rungs, respectively, p < 0.001).

In assessing the change in intent from pre- to post-intervention, there was a significant interaction between baseline intent to decrease regular use of OTC NSAIDs and group (β = −0.16, t(913)= −3.88; p < 0.001); i.e., a rate dependence) such that, compared with CONTROL, VIDEO participants with a lower baseline intent to decrease regular use of OTC NSAIDs showed greater increases in intent following the intervention vs. those exhibiting higher intent to decrease regular use of OTC NSAIDs at baseline (Fig 2A). In other words, VIDEO had a greater influence on intent to decrease regular use of OTC NSAIDs immediately post-intervention for participants with lower vs. higher baseline intent and this influence was significantly higher in VIDEO compared with CONTROL.

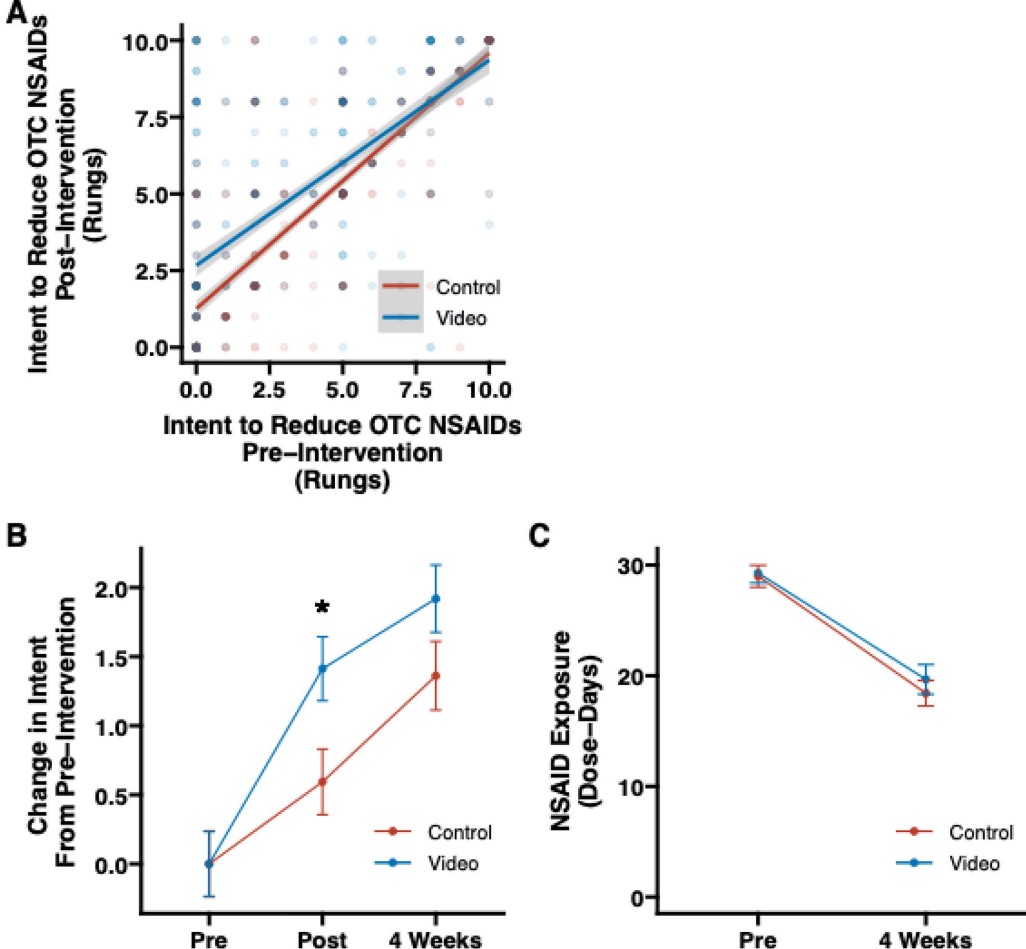

**Fig 2. Intent to decrease use of OTC NSAIDs and NSAID exposure.** (A) Scatter plot of the group effect on intent to decrease use of OTC NSAIDs at pre-intervention and post-intervention. Color represents group assignment (CONTROL, VIDEO) and saturation of a point indicates more observations are present at that point (n = 917). (B) Change in intent to decrease regular use of OTC NSAIDs from baseline through 4-week follow-up by group (n = 425). (C) Change in NSAID Exposure from baseline through 4-week follow-up up by group (n = 425). OTC=Over-the-counter, NSAID = non steroidal anti-inflammatory drug.

**Intent to decrease regular use of OTC NSAIDs 4 weeks post-intervention.** After 4 weeks, intent to decrease use of OTC NSAIDs was similar in VIDEO and CONTROL (6.02 (SD: 3.64) vs. 5.85 (SD: 3.53) rungs, respectively, p = 0.61). Participants in both groups reported a rise in their intent to decrease use of OTC NSAIDs but the change in intent was not significantly different between groups (1.92 (SD: 4.41) vs. 1.36 (SD: 3.46) rungs for VIDEO and CONTROL, respectively, p = 0.150) as shown in **Fig 2B**. In assessing the change in intent to decrease regular use of OTC NSAIDs from pre-intervention to the 4-week follow-up, there was a significant interaction between pre-intervention intent and group (β = −0.28, t(421) = −2.94; p = 0.003) such that the influence of baseline intent levels (i.e., lower levels at baseline = greater change in intent at the 4-week follow-up) was greater for VIDEO vs. CONTROL participants.

**NSAID exposure at 4 weeks post-intervention.** There was also no significant difference between VIDEO and CONTROL in NSAID Exposure at the 4-week Follow-Up assessment (19.67 (SD: 19.60) and 18.42 (SD: 15.72) dose-days, respectively; p = 0.48). From pre-intervention to 4 weeks post-intervention, OTC NSAID Exposure decreased by 9.61 (SD: 14.26) dose-days in VIDEO (a 32.8% decrease) and 10.58 (SD: 16.10) dose-days in CONTROL (a 36.5% decrease), with no significant difference between groups (p = 0.46), as seen in **Fig 2C**.

### Secondary outcomes

**COM-B domain scores at 4 weeks post-intervention.** Overall Capability, Opportunity, and Motivation domain scores were 7.58 (SD: 1.99), 6.49 (SD: 2.78), and 6.56 (SD: 3.28), respectively, on the 11-point Likert scale pre-intervention and 7.92 (SD: 1.88), 6.73 (SD: 2.70), and 6.77 (SD: 3.30), respectively, at the 4-week follow-up assessment (**Table 2**). Each domain increased 2.7% to 6.6% in both groups. There were no significant group differences in Capability, Opportunity, and Motivation at either time point (**Table 2**), nor in the change in Capability, Opportunity, and Motivation from pre- to post-intervention.

**Pain measures at 4 weeks post-intervention.** Current, recent average, and recent worst pain were rated as 2.45 (0.98) -> 2.51 (1.00), 3.00 (0.80) -> 2.88 (0.87), and 3.81 (0.96) -> 3.61 (1.00), respectively on the 5-point Likert scale, with no significant group differences (**Table 2**). Ratings of current pain increased 0.40% from pre- to post-intervention in VIDEO and 4.6% in CONTROL, while ratings of average recent pain decreased by 3.6% from pre- to post-intervention in VIDEO and 4.3% in CONTROL, with no differences between groups (p = 0.09 and 0.39, respectively). The change in ratings of worst recent pain from pre- to post-intervention (6.9% decrease in VIDEO vs. 3.2% decrease in CONTROL) approached statistical significance (p = 0.05).

**COM-B domain ratings as moderators of intent to decrease use of OTC NSAIDs and self-reported OTC NSAID exposure.** Two COM-B domains (Opportunity and Motivation) moderated the relationship between group and intent to decrease OTC NSAIDs immediately post-intervention (Opportunity: β = 0.23, t(419) = 2.90; p = 0.004; Motivation: β = 0.19, t(418) = 2.75; p = 0.006) such that VIDEO participants with higher baseline Opportunity and Motivation ratings reported higher intent than CONTROL participants (**Fig 3A** and **3B**). One domain (Opportunity) moderated the relationship between group and intent to decrease OTC NSAIDs 4 weeks post-intervention (β = 0.30, t(419) = 2.13; p = −0.03) such that VIDEO participants with higher baseline Opportunity ratings reported higher intent than CONTROL participants (**Fig 3C**). None of the three COM-B domains moderated the relationship between group and change in total NSAID Exposure.

**Pain measures as moderators of intent to decrease use of OTC NSAIDs and self-reported OTC NSAID exposure.** None of the pain measures moderated the relationship between group and intent to decrease use of OTC NSAIDs immediately or 4 weeks post-intervention, nor the relationship between group and self-reported NSAID Exposure at 4 weeks.

## Discussion

The National Kidney Foundation, American Heart Association, American Geriatrics Society, Choosing Wisely campaign [8–10], and others recommend that individuals with CKD, HF, and/or HTN avoid the regular use of NSAIDs due to the

**Table 2. Results for the primary and secondary outcomes in regular over-the-counter NSAID users randomized to intervention and control groups at baseline and 4 weeks after the intervention, who completed the follow-up assessment.**

| | VIDEO Mean (SD) (n = 223) | CONTROL Mean (SD) (n = 202) | p-value |
|---|---|---|---|
| **Intent to Decrease Use of OTC* NSAIDs[†][‡](Contemplation Ladder rungs)** | | | |
| Baseline | 4.10 (3.49) | 4.49 (3.39) | 0.25 |
| Post-Intervention | 5.52 (3.46) | 5.08 (3.37) | 0.19 |
| 4-week Follow-Up | 6.02 (3.64) | 5.85 (3.53) | 0.61 |
| **NSAID Exposure[§](Dose- Days)** | | | |
| Baseline | 29.28 (13.33) | 29.00 (14.05) | 0.83 |
| 4-Week Follow Up | 19.67 (19.60) | 18.42 (15.72) | 0.48 |
| **Pain[∥]** | | | |
| *Current* | | | |
| Baseline | 2.52 (0.96) | 2.38 (0.99) | 0.12 |
| 4-Week Follow Up | 2.53 (0.97) | 2.49 (1.03) | 0.66 |
| *Average* | | | |
| Baseline | 3.00 (0.81) | 3.00 (0.79) | 0.81 |
| 4-week Follow Up | 2.89 (0.89) | 2.87 (0.86) | 0.77 |
| *Worst* | | | |
| Baseline | 3.88 (0.96) | 3.73 (0.95) | 0.06 |
| 4-Week Follow Up | 3.61 (1.03) | 3.61 (0.97) | 0.78 |
| **COM-B[**]** | | | |
| *Capability* | | | |
| Baseline | 7.74 (2.01) | 7.41 (1.97) | 0.09 |
| 4-week Follow Up | 7.95 (1.93) | 7.90 (1.82) | 0.76 |
| *Opportunity* | | | |
| Baseline | 6.73 (2.73) | 6.23 (2.80) | 0.06 |
| 4-week Follow Up | 6.96 (2.62) | 6.49 (2.77) | 0.07 |
| *Motivation* | | | |
| Baseline | 6.56 (3.37) | 6.53 (3.20) | 0.94 |
| 4-week Follow Up | 6.78 (3.33) | 6.73 (3.29) | 0.86 |

* OTC = over-the-counter

[†]NSAIDs = nonsteroidal anti-inflammatory drugs

[‡]Intent to Decrease use of OTC NSAIDs was evaluating using the 11-point Contemplation Ladder where a higher rating indicates greater intention.

[§] NSAID Exposure was evaluated via the summative frequency of all four NSAID category questions: ibuprofen or naproxen, aspirin, headache tablets and powders, other NSAIDs such as some cold & sinus medications and some menstrual pain medications.

[∥] Pain was evaluated using a 5-point Likert scale addressing Current, Average, and Worst recent pain.

[**] COM-B was evaluated via 7 questions using an 11-point Likert scale.

potential risks. Despite these recommendations, over 40% of individuals with CKD, HF, and/or HTN who screened for the present study self-identified as regular OTC NSAID users. Viewing a brief, novel educational video resulted in a 32% increase in intent to decrease OTC NSAIDs immediately after viewing, an increase maintained at 4 weeks post-intervention. Interestingly, the control condition (FDA Drug Facts label for OTC NSAIDs) was also associated with a 13% increase in intent to decrease OTC NSAIDs. Both conditions facilitated a significant decrease in self-reported OTC NSAID use (approximately 10 dose-days per month). These findings suggest that a brief, low-burden intervention can improve OTC NSAID safety in patients with chronic health conditions.

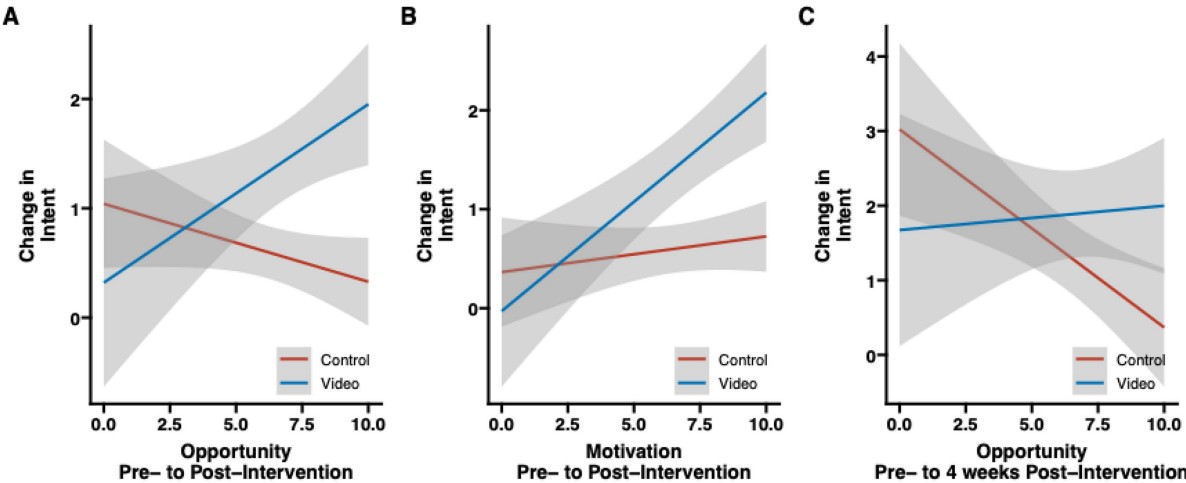

**Fig 3. Moderating analysis for Opportunity and Motivation.** (A) Moderating effect of Opportunity between group and intent to decrease OTC NSAIDs immediately post-intervention. (B) Moderating effect of Motivation between group and intent to decrease OTC NSAIDs immediately post-intervention. (C) Moderating effect of Opportunity between group and intent to decrease OTC NSAIDs 4 weeks post-intervention. OTC=Over-the-counter, NSAID = non steroidal anti-inflammatory drug.

Given the broad availability of OTC NSAIDs in the US, the prevalence of CKD, HF, and/or HTN (50% of Americans), and high rates of chronic pain in patients with chronic disease [30], dissemination of an effective, low-burden educational initiative has substantial implications for improving population health through facilitation of self-moderated behavior change. As many NSAID-related adverse effects are dose-dependent, any reduction in OTC NSAID use may benefit patients with CKD, HF, and/or HTN [7]. In the present study, the video was associated with a meaningful improvement in intent to decrease OTC NSAIDs, with nearly a 2-rung increase on the contemplation ladder and a 10 dose-day per month reduction in self-reported OTC NSAID use after 4-weeks, changes that significantly greater than observed in previously following a single intervention [13,14,31]. Movement up the contemplation ladder reflects a shift to a higher stage in the transtheoretical model of change, such as moving from precontemplation to contemplation, or contemplation to preparation, which is associated with a greater likelihood of sustained behavioral change [20]. It is encouraging that no reciprocal increase in pain levels was associated with reductions in OTC NSAIDs.

Prior studies have described interventions effective in improving knowledge about OTC NSAID risks, which addresses the capability domain of COM-B. Our study is among the first to demonstrate behavior change following a single intervention. The success of the intervention could be attributed, at least in part, to the COM-B-informed video design, as the domains of opportunity and motivation positively moderated the effectiveness of the video. The video was most effective for participants with low initial intent who reported higher levels of opportunity and motivation.

It is noteworthy that our control group, which viewed the FDA OTC NSAIDs Drug Facts label, also experienced improvement in intent to decrease OTC NSAID use and in self-reported use at 4 weeks. Others have reported barriers to public comprehension of OTC drug facts labels [32,33], so the enlarged version viewed by the control group may have benefitted participants by improving access to the content. Further research is needed to understand how to optimize the FDA's Drug Facts label within video-based educational interventions to de-implement high-risk NSAIDs.

This study has some limitations. First, our control group viewed the FDA OTC NSAIDs Drug Facts label rather than receiving no intervention. Our research team and ethics committee believed the label comparison was more appropriate than providing no education about this important topic. Thus, we are limited in our ability to assess the impact of the

video intervention compared with standard practice (no education). Second, we assessed self-reported OTC NSAIDs use. Although there are limitations to patient-reported medication use data, we employed a previously used instrument and received similar reports about the frequency of high-risk OTC NSAID to others [5,27]. We acknowledge that the instrument may not have been sensitive to detecting minor changes in NSAID use. Third, we did not evaluate prescription NSAIDs or other pain therapies that the patient might have been using, which may have impacted OTC NSAIDs use. Fourth, we did lose a substantial number of participants to follow-up. However, there were no significant differences in demographic and clinical factors for participants who completed the intervention and immediate post-intervention assessment and those who completed the 4 week post-intervention assessment. Finally, the generalizability of our findings may be limited by lack of clinical and demographic diversity in our sample.

## Conclusion

In the US, OTC NSAIDs are broadly available and commonly used by high-risk patients, including those with CKD, HF, and HTN. The present investigation showed that viewing a brief COM-B-informed educational video or the FDA OTC NSAIDs Drug Facts label resulted in a significant rise in participants' intent to decrease use of OTC NSAIDs and a 10 dose-day per month decrease in self-reported use of OTC NSAIDs 4 weeks post-intervention with no reported change in current or recent pain levels. The video was significantly more effective than the label for participants with the lowest baseline intent to decrease use of OTC NSAIDs. These results suggest that an electronically-administered intervention is a low-burden, reproducible approach with potential for improving medication safety and population health for patients with CKD, HF, and/or HTN. Further research is needed to confirm and contextualize the moderating role of the COM-B domains of opportunity and motivation on effectiveness of such interventions.

## Supporting information

**S1 File. Screening survey.**
(DOCX)

**S2 File. ICD-10 codes.**
(DOCX)

**S3 File. Randomization process.**
(DOCX)

**S4 File. Video.** The video and related images are property of the Virginia Tech Carilion School of Medicine. They are reprinted and made available in this publication under a CC BY license, with permission from the Virginia Tech Carilion School of Medicine, original copyright 2024.
(DOCX)

**S5 File. Control/Label.**
(DOCX)

**S6 File. Full survey questions.**
(DOCX)

**S7 File. CONSORT diagram.** *OTC= over-the-counter, †NSAID = nonsteroidal anti-inflammatory drug, ‡Excluded: 132 participants were excluded due to: (a) reported low-dose aspirin use for cardiovascular disease prevention rather than pain management (n = 63); (b) misclassified as eligible to participate due to administrative error (n = 66); and (c) duplicate responses (n = 3).
(DOCX)

## Acknowledgments

The authors acknowledge Mr. Ryan Anderson from the Virginia Tech Carilion School of Medicine and the Virginia Tech Technology-Enhanced Learning and Online Strategies for their contributions to video design and production, Mr. Emma Gahima Oyese from Carilion Clinic for survey support, and Ms. Helga Morrow and Dr. Jasmine Jones for serving as actors in the study video.

## Author contributions

**Conceptualization:** Matthew G. Vinson, John W. Epling, Michelle S. Rockwell.

**Data curation:** Mamata R. Tokala.

**Formal analysis:** Matthew G. Vinson, Allison N. Tegge, Martha M. Tenzer.

**Funding acquisition:** Michelle S. Rockwell.

**Investigation:** Matthew G. Vinson, John W. Epling, Michelle S. Rockwell.

**Methodology:** Jeffrey S. Stein.

**Project administration:** Cara R. Spivey.

**Resources:** Cara R. Spivey.

**Software:** Mamata R. Tokala.

**Supervision:** Martha M. Tenzer, John W. Epling, Michelle S. Rockwell.

**Visualization:** Allison N. Tegge, Jeffrey S. Stein.

**Writing – original draft:** Matthew G. Vinson.

**Writing – review & editing:** Allison N. Tegge, Mamata R. Tokala, Cara R. Spivey, Martha M. Tenzer, Jeffrey S. Stein, John W. Epling, Michelle S. Rockwell.

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
