## [Decision Letter · Decision Letter 0]

24 Jul 2025

PONE-D-25-15633
A brief, theory-driven patient education video reduces high-risk over-the-counter nonsteroidal anti-inflammatory drug (NSAID) use
PLOS ONE

Dear Dr. Rockwell,

Thank you for submitting your manuscript to PLOS ONE. After careful consideration, we feel that it has merit but does not fully meet PLOS ONE’s publication criteria as it currently stands. Therefore, we invite you to submit a revised version of the manuscript that addresses the points raised during the review process.

We look forward to receiving your revised manuscript.

Kind regards,

Deema Jaber, Ph.D.

Academic Editor

PLOS ONE

Journal Requirements:

“This research was funded, in part, by a collaborative grant from the Virginia Tech Libraries.”

“This research was supported by a collaborative research grant from the Virginia Tech Libraries.”

“This research was funded, in part, by a collaborative grant from the Virginia Tech Libraries.”

5. We note that Supplementary figure in your submission contain copyrighted images. All PLOS content is published under the Creative Commons Attribution License (CC BY 4.0), which means that the manuscript, images, and Supporting Information files will be freely available online, and any third party is permitted to access, download, copy, distribute, and use these materials in any way, even commercially, with proper attribution. For more information, see our copyright guidelines: http://journals.plos.org/plosone/s/licenses-and-copyright.

1. You may seek permission from the original copyright holder of Supplementary figure to publish the content specifically under the CC BY 4.0 license.

Additional Editor Comments:

Dear Rockwell,

Thank you for submitting your manuscript entitled "A brief, theory-driven patient education video reduces high-risk over-the-counter nonsteroidal anti-inflammatory drug (NSAID) use, (Manuscript Number PONE-D-25-15633)" to Plos One. We have now received comments from the reviewers, and I have carefully considered their evaluations along with my own assessment.

I am pleased to inform you that your manuscript has been judged to be of interest and merit for publication in our journal. However, the reviewers have suggested minor revisions to improve clarity and address a few remaining concerns.

Please refer to the enclosed reviewer comments for specific feedback. We kindly ask that you:

Revise the manuscript in response to the reviewers’ suggestions.

Provide a point-by-point response letter detailing how each comment has been addressed.

Highlight all changes made in the revised manuscript (e.g., using track changes or colored text).

To ensure timely processing of your submission, we request that you resubmit the revised manuscript within one month from the date of this letter.

We appreciate your contribution to Plos One, and we look forward to receiving your revised manuscript.

Kind regards,

Deema Jaber

Academic Editor

Plos One

Reviewers' comments:

Reviewer's Responses to Questions

**Comments to the Author**

1. Is the manuscript technically sound, and do the data support the conclusions?

Reviewer #1: Yes

Reviewer #2: Yes

2. Has the statistical analysis been performed appropriately and rigorously? 

Reviewer #1: Yes

Reviewer #2: Yes

3. Have the authors made all data underlying the findings in their manuscript fully available?

Reviewer #1: Yes

Reviewer #2: Yes

4. Is the manuscript presented in an intelligible fashion and written in standard English?

Reviewer #1: Yes

Reviewer #2: Yes

5. Review Comments to the Author

Reviewer #1: Thank you for your invitation to revise the article entitled:

A brief, theory-driven patient education video reduces high-risk over-the-counter nonsteroidal anti-inflammatory drug (NSAID) use.

I have few minor comments:

1- We need reference in line 85

2- I suggest to add (in patients with CKD, HF, and/or HTN) in the last of the title to be; A brief, theory-driven patient education video reduces high-risk over-the-counter nonsteroidal anti-inflammatory drug (NSAID) use in patients with CKD, HF, and/or HTN.

3- In discussion chapter line 351; we need a reference.

4- Line 254; “In assessing the change in intent from pre- to post-intervention,…..” where is the same assessment for control group?

5- In line 299; the authors say “After 4 weeks, intent to decrease use of OTC NSAIDs was similar in VIDEO and CONTROL” …how come while there wase a significant difference in intent between the two groups immediately after intervention (you say), the change in intent was not significantly different between groups after 4 weeks (you say) and there was no significant difference between pre and post 4 weeks period in both groups (you say)? It does make sense if there was a significant difference in intent between the two groups after two weeks also.

6- In conclusion part the authors didn’t mention the comparison between the two studied groups as I think overall differences wasn’t found. The control group as the authors say “There was also no significant difference between VIDEO and CONTROL in NSAID Exposure at the 4-week Follow-Up assessment (19.67 (SD: 19.60) and 18.42 (SD: 15.72) dose-days, respectively; p = 0.48).” and also say “From pre-intervention to 4 weeks post-intervention, OTC NSAID Exposure decreased by 9.61 (SD: 14.26) dose-days in VIDEO (a 32.8% decrease) and 10.58 (SD: 16.10) dose-days in CONTROL (a 36.5% decrease), with no significant difference between groups (p = 0.46)”. this means (image of the Food and Drug Administration (FDA) Drug Facts label for OTC NSAIDs) that digested by control group is as effective as video (in the test group) in reducing NSAIDs exposure after the 4 weeks period of the study. The authors frankly mentioned that throughout the context of the article but, they should clarify that in the conclusion part.

7- Overall the article is good and concised.

Reviewer #2: This manuscript discusses relevant issue which regulates around patients with CKD, DM and hypertension. The abstract describes the conduct of the study in detail. There are several missing points that could be added to the abstract for better clarity i.e. state briefly guidelines that highlight the avoidance of NSAIDs use among CKD patients, specifications on NSAIDs consumption (duration, how long, dose etc)/definiton of regular NSAID use. A brief sentence on possible reason as to why control group has greater decrease in exposure to NSAIDs could be added for better study summary.

Methodology (Study design): Specific details on the date of RCT can perhaps be stated as "between January to September 2023". Similarly for the description under setting and participants, the specific date mentioned can be omitted.

Line 109 to 118- The setting for data collection was not fully described. It is understood that background data is obtained from the EHR and emails were sent to patients to screen for eligible patients. What happened after that, patients come in for face to face interview/intervention? Appreciate if you could put in more details to help the reader understand the data collection processes. Even figure 1 did not detail out the study flow.

Line 138 " small effect size and statistical significance at p<0.05, assuming a 50%...."

I like the way the video intervention was described. Very clear and concise.

Line 196, under outcome measures, the calculation stated was not quite clear where the average NSAID exposure was calculated as 5.5 dose-days per week X 4 weeks for administration of 3-5 days of ibuprofen and 1-2 days of aspirin. It would be helpful if it could be described using a formula.

Conclusion- The conclusion should state that there was no significant difference between those who underwent video intervention vs the control group rather than hiding the actual outcome. This could be misleading. The manuscript clearly stated the limitations of the study, therefore, highlighting the actual outcome will be more appropriate. Perhaps, a brief statement on future work can be included to give a complete closure to the well-written manuscript.

Figure 1: I think the title 'General Study Design' is not suitable as the diagram is not showing study design.

6. PLOS authors have the option to publish the peer review history of their article (what does this mean?). If published, this will include your full peer review and any attached files.

Reviewer #1: **Yes: **Ahmed Amin Ali

Reviewer #2: **Yes: **Aina Yazrin Ali Nasiruddin

---

## [Author Response · Author response to Decision Letter 1]

5 Oct 2025

Please see the file uploaded with a point-by-point response to the editor and reviewers.

---

## [Editor Report · Decision Letter 1]

12 Oct 2025

A brief, theory-driven patient education video reduces high-risk over-the-counter nonsteroidal anti-inflammatory drug (NSAID) use

PONE-D-25-15633R1

Dear Dr. %Rockwell%,

We’re pleased to inform you that your manuscript has been judged scientifically suitable for publication and will be formally accepted for publication once it meets all outstanding technical requirements.

Kind regards,

Deema Jaber, Ph.D.

Academic Editor

PLOS ONE
---

## [Editor Report · Acceptance letter]

PONE-D-25-15633R1

PLOS ONE

Dear Dr. Rockwell,

I'm pleased to inform you that your manuscript has been deemed suitable for publication in PLOS ONE. Congratulations! Your manuscript is now being handed over to our production team.

Kind regards,

on behalf of

Dr. Deema Jaber

Academic Editor

PLOS ONE